# Multi-Task Offloading Based on Optimal Stopping Theory in Edge Computing Empowered Internet of Vehicles

**DOI:** 10.3390/e24060814

**Published:** 2022-06-11

**Authors:** Liting Mu, Bin Ge, Chenxing Xia, Cai Wu

**Affiliations:** 1College of Computer Science and Engineering, Anhui University of Science and Technology, Huainan 232001, China; ltingmu@163.com (L.M.); cxxia@aust.edu.cn (C.X.); wuc17855370929@163.com (C.W.); 2Institute of Energy, Hefei Comprehensive National Science Center, Hefei 230031, China

**Keywords:** mobile edge computing, computational offloading, optimal stopping theory, structured tasks, time optimization, sequential decision making

## Abstract

Vehicular edge computing is a new computing paradigm. By introducing edge computing into the Internet of Vehicles (IoV), service providers are able to serve users with low-latency services, as edge computing deploys resources (e.g., computation, storage, and bandwidth) at the side close to the IoV users. When mobile nodes are moving and generating structured tasks, they can connect with the roadside units (RSUs) and then choose a proper time and several suitable Mobile Edge Computing (MEC) servers to offload the tasks. However, how to offload tasks in sequence efficiently is challenging. In response to this problem, in this paper, we propose a time-optimized, multi-task-offloading model adopting the principles of Optimal Stopping Theory (OST) with the objective of maximizing the probability of offloading to the optimal servers. When the server utilization is close to uniformly distributed, we propose another OST-based model with the objective of minimizing the total offloading delay. The proposed models are experimentally compared and evaluated with related OST models using simulated data sets and real data sets, and sensitivity analysis is performed. The results show that the proposed offloading models can be efficiently implemented in the mobile nodes and significantly reduce the total expected processing time of the tasks.

## 1. Introduction

The Internet of Vehicles (IoV) is an emerging concept in intelligent transportation systems that aims to improve traffic safety and passenger comfort through integration with the Internet of Things (IoT), and it is an important implementation of the IoT [1]. Based on Vehicle to Everything (V2X) technology, the IoV connects vehicles, roadside units (RSUs), and service providers into a whole organic network, enabling all-round communication between them [2]. Smart vehicles in the IoV can communicate via V2X. Specifically, smart vehicles can share information with other vehicles through Vehicle to Vehicle (V2V) communication; thus, it can obtain a wider view of the road condition information shared by surrounding vehicles and greatly reduce the traffic accidents caused by blind spots [3].

As the number of vehicles on the road continues to increase, the IoV continues to evolve, smart vehicles accounting for an increasing share of Internet-connected devices. In the IoV paradigm, smart vehicles are equipped with computing units and communication technologies that provide services such as intelligent control, traffic management and interactive applications for vehicles. The edge-side computing architecture for autonomous vehicles (AVs) relies on the communication infrastructure and services provided by the edge-cloud collaboration and the (Long-Term Evolution/5th Generation Mobile Communication Technology) LTE/5G. The edge side mainly refers to on-board edge computing units, RSUs, Mobile Edge Computing (MEC) servers, etc. AVs are equipped with a large number of sensors that collect data for different types of traffic systems and navigation applications, etc. [4]. Currently, each AV is equipped with more than 60 to 100 electronic control units to support various functions such as communication, engine, dashboard, seat control, entertainment, etc. AVs generate a large amount of data in real time while driving; for example, AVs will generate and consume about 40 TB of data for every eight hours of driving (e.g., a city’s High-Definition (HD) map is about 1.5TB) [5]. Although AVs have small-scale computational and storage resources, the computational capacity of vehicle terminals is relatively limited due to the huge computational tasks and the very high demand for real-time system response; e.g., in hazardous situations, the response time of the vehicle braking system is directly related to the safety of the vehicle, passengers, and the road, so they rely on other computational resources [6]. The MEC servers in the rear seat unit of AVs can play an important role in improving the performance of mobile vehicle terminals [7].

Since MEC servers operate at the edge of the radio access network, connecting to RSUs and performing transmission tasks with their help, their service area may be limited by the radio coverage of the RSUs. Due to the high mobility of the nodes, moving vehicles may pass through multiple RSUs and MEC servers during task offloading and can offload their computational tasks to any MEC server they have access to that provides computational resources for the tasks offloaded by the mobile nodes [8], as shown in Figure 1. The load on the MEC servers varies greatly from moment to moment; sometimes, there are a large number of users using the same server at the same time, causing a high load situation, while at other times, only a few users are connected to the server and the server load is low [9]. Therefore, the key to the problem is how the mobile node defines the offloading decision by which it selects the appropriate edge servers to offload the computational tasks so that the total computational latency is minimized [10].

A centralized offloading strategy has been used in some studies, where vehicles can request information about MEC servers available for offloading along the road from a centralized server located at a higher level [11]. The centralized server is connected to the MEC servers through a wired network. However, this centralized architecture is not visible, and as the number of vehicles on the road continues to increase, the centralization of road information puts a higher load on the network [9]. Offloading strategies based on Vehicle-to-Vehicle communication (V2V) architectures are discussed in other studies, where vehicles can send tasks to the MEC server through other vehicles, each of which can act as an intermediate node for data delivery [4]. This solution requires the presence of other vehicles, but there is no guarantee that other vehicles will be present. Therefore, considering the scenario that if the mobile node only knows the MEC server that is being observed sequentially and has no global information about the candidate MEC server or only local information, how should it choose the optimal MEC server for offloading?

Such problems are known as Optimal Stopping Theory (OST) problems. A basic question about OST is that nodes autonomously decide when to act based on sequentially observed random variables with the goal of increasing expected benefits or decreasing expected costs [12]. The Secretary Problem (SP), the House Sale (HS) problem or the Fair Coin Problem are some well-known OST problems [12]. Single-node OST offloading problems have been studied, and in this paper, we attempt to study OST-based offloading problems for structured tasks with multiple subtasks and provide lightweight local algorithms that can be implemented in mobile nodes (vehicles or smartphones) so that mobile nodes can make offloading decisions independently.

The remainder of this paper is organized as follows: we provide a summary of the related work and outline our contributions in Section 2, while details of the system model and the problem formulation are described in Section 3. The OST-based task offloading models are described in Section 4.1 and Section 4.2. Performance evaluation is provided in Section 5. Finally, Section 6 concludes the paper and outlines future research directions.

## 2. Related Work and Contributions

### 2.1. Related Work

In dealing with the issue of how to offload computational tasks and data to edge nodes, various pieces of research have emphasized whether data or tasks should be offloaded to servers or computed locally. The offloading decision has two main goals: minimization of computational delay and energy consumption.

Ref. [13] proposed the ST-CODA algorithm for computing offloading decisions based on space and time in this paper to design mobile devices’ decisions in terms of time and location for offloading tasks by considering computing nodes and transmission costs in heterogeneous networks supported by edge clouds, where the time decision refers to postponing the offloading decision until a low-cost network, such as a wireless network, is found. Unlike ST-CODA, the approach studied in this paper will only offload tasks to edge servers and not further to the cloud, and the decision is deferred until a less loaded server is found.

Ref. [14] used Device-to-Device (D2D) communication technology to establish a direct link between mobile devices (MDs) and proposed a Secure and Energy-aware Collaborative Task Offloading for D2D communication (Sec2D) considering the data security. A new security model is constructed in terms of CPU cores, CPU frequency, and data size to measure the secure workload on heterogeneous MDs with guaranteed data security, a collaborative task offloading problem is proposed to minimize the time-averaged delay and energy consumption of MDs, and the Lyapunov optimization framework is applied to achieve online decision making. Greedy approach and optimal approach with different time complexity are proposed to deal with the generated mixed-integer linear programming (MILP) problem. The D2D technique mentioned in the paper is similar to the V2V communication architecture in the vehicular network, which can reduce the communication delay and improve the network capacity of the underlying wireless network.

In [15], OST was employed in order to obtain a good balance between the benefit of selecting the best edge device and the cumulative cost of deep resource detection. The authors tried to enhance the OST-based model by using a hierarchical learning mechanism to define the OST threshold and the sequence of edge nodes used for offloading. However, as it implements deep neural networks and deep Q-networks, it leads to a significant increase in the computational overhead and battery consumption of mobile nodes. Moreover, in their application, it is assumed that once the task is generated, the mobile node will have a list of edge devices, and then, the mobile node will define which edge node strikes a good balance between detection cost and execution delay of the task in advance.

In [16,17], the authors proposed a set of lightweight sequential decision models using the OST principle. In particular, Delay-Tolerant Offloading (DTO) decisions in the MEC environment are proposed, in which the unloading decision should be made within a predefined time horizon. This can be considered as a finite time horizon OST problem. This has been further investigated in [18], and two OST-based models have been introduced. The first one is the Best Choice Problem-based optimal task offloading policy (BCP), where the objective is to maximize the probability of offloading to the best edge server. The advantage of this model is that task offloading decisions can be made independently by mobile nodes, and it only needs to provide the number of observations (the number of edge nodes). Another one is the Cost-based Optimal Task Offloading Policy (COT), which enhances the BCP model by further considering the cost per observation of the server and obtaining more metric information about the performance. In [19], the authors summarized this series of OST-based offloading models, introduced the concept of contextual data quality awareness, proposed a timeliness function, injected offloading decisions, proposed an OST-based optimal selection method, and provided detailed evaluation and analysis by modeling and experimenting the application of OST models in MEC environments.

In [20], the authors considered the possible existence of the situation of multi-tasks and structured tasks offloading. A system model for a multi-user, multi-offload point and structured task environment is first developed to formalize the offload decision problem in this environment as a cost minimization problem, and then, a backtracking-based approach is designed to obtain its exact solution. To reduce the complexity, a method based on an improved genetic algorithm and a method based on a greedy strategy are designed. Finally, the three methods are verified and compared in terms of total cost and resource utilization at the unloading point and execution time for all users.

In this work, we consider the case of dividing the tructured tasks into multiple subtasks [20]. To solve the multi-task sequential offloading problem, inspired by the work of [19], we design distributed multi-task offloading model based on OST. The offloading models should be as lightweight as possible to facilitate local execution by mobile nodes. We also introduce the time-optimized function to ensure the timeliness of the offloading process. Additionally, we provide a comparative evaluation of all the OST-based models found in the literature with other offloading methods using simulation-based evaluation and real data set evaluation.

### 2.2. Contribution

In summary, our contributions are:We consider the multi-task offloading problem based on OST, where structured tasks are divided into multiple subtasks and offloaded to the MEC servers with the goal of minimizing the total task processing time.We propose a time-optimized function to simulate the situation where a mobile node needs to perform offloading as far as possible before the task data become obsolete in a realistic scenario and inject it into the decision model.We propose a time-optimized multi-task offloading model based on OST aiming at maximizing the probability of offloading to the optimal servers for general distribution scenarios.We propose a better time-optimized multi-task offloading model based on OST with the objective of minimizing the total task processing time for some realistic road scenarios where the CPU utilization may have an approximately uniform distribution.We provide an experimental comparison, performance evaluation and sensitivity analysis of the models in this paper and a series of models in other research using simulated and real data sets.

## 3. System Model and Problem Description

Consider a vehicle networking application scenario studied in [4,10]: mobile nodes travel on the road with RSUs and base stations deployed on the roadside, where a limited set of MEC servers with storage units and computation units are deployed [21]. Vehicles traveling on the road generate computational tasks and try to offload them to the MEC servers along the road. As user requirements become more complex, individual tasks gradually fail to meet user needs, and mobile nodes may generate a large number of structured tasks. For example, a structured task that is generated to meet business travel needs can be divided into three subtasks: weather forecasting ($t_{1}$), flight reservation ($t_{2}$), and hotel reservation ($t_{3}$). $t_{2}$ and $t_{3}$ are independent of each other and thus can be executed in parallel, but they all depend on the results provided by $t_{1}$ [22]. So, the three subtasks can be executed in the order of $t_{1}$, $t_{2}$, and $t_{3}$ for offloading, and thus, they can be offloaded in chunks by dividing the structured task into multiple subtasks [20].

The MEC server can operate at the edge of the network with the help of the RSU, so its communication range will be limited by the RSU communication range [5]. Since the input data for the vast majority of tasks are much larger than the output data of the computation results, to ensure the continuity of task completion, it can be assumed that there are mobility management entities in the server that implement mobility management algorithms, such as path selection, power control algorithms [23,24] or as the prediction model in [4]. If the node needs to obtain some results from a server other than the one selected for offloading this task, and the mobile node is outside the range of this MEC server, the selected MEC server should use a high-bandwidth wired connection to transmit the task results to the node through the next MEC server [25], thus solving the problem of continuity in the transmission of computational results after the vehicle is out of communication range of the MEC server.

The mobile node observes the MEC servers along the route in order at this point, and $X_{i}$ is the random variable of the *i*th server processing time observed. When a node generates tasks locally and needs to offload, the node can observe *n* MEC servers; the mobile node uses the network and server analyzer to check each MEC server it passes and obtains the *X* values. Assuming that *K* blocks of tasks need to be unloaded, in *n* observation durations, then *K* unloading decisions need to be made.

The current goal is to optimize the decision to offload multiple targets to available servers. Overall, the offloading objectives are twofold: (1) maximize the probability of offloading to *K* optimal servers; (2) minimize the expected value of the total *X* for *K* tasks. We first propose an offloading model that is generalizable to the distribution of *X* for the first objective, and then, we propose a better offloading model with the second objective (minimizing the expected value of the total *X* for *K* tasks) for certain road scenarios where there may be a uniform distribution of *X*. Table 1 provides the key notations used in this paper.

## 4. Computational Model

### 4.1. General Model

Consider first the OST problem model in the K=1 case, where the goal is to maximize the probability of offloading to the best node, thus achieving the effect that the total processing time of the system is as small as possible. In the previous BCP model [18], the offloading scenario considered in this paper can be equated to the scenario discussed in this paper K=1 where the mobile nodes know in advance the number of MEC servers that are available for offloading as n(n>0,n≥K). The value of the specific *n* can be defined by the user or estimated by the application based on the deadlines of the tasks to be offloaded. Each node can only check the servers in a sequential and random order and will rank them relatively based on the previously observed servers. At each observation, the node should decide whether to select the current candidate server, and once selected, it cannot revoke its decision. The node should maximize the probability of selecting the best candidate among *n* probability of choosing the best among the candidates. This is a best choice problem [12]. The offloading rules in BCP are described as follows.

Call the first *i* server as the candidate server, the i=1,…,n, assuming that it is the best in terms of $X_{i}$. We define a positive integer $r_{n}$ in {1,…,n}, which is defined as:(1)$$r_{n}=min\left\{r\geq 1:\frac{1}{r}+\frac{1}{r+1}+\ldots +\frac{1}{n-1}\leq 1\right\}.$$

For n≥2, according to BCP, the best strategy is to reject the first $r_{n}-1$ server and then select the first candidate server (if any) to offload the task. The maximum probability of selecting the best candidate for BCP in (1) is given by the following equation:(2)$$P_{n}\left(r_{n}\right)=\frac{r_{n}-1}{n}\sum_{i=r_{n}}^{n}\frac{1}{i-1}.$$

For smaller values of *n*, the optimal stop node $r_{n}$ can be calculated using (1). When n→∞, we obtain the well-known secretary problem, where the optimal probability tends to $\frac{1}{e}$ [26]. Figure 2 shows the values of the probability of offloading to the best candidate and $R_{n}$ for different values of *n*. It can be seen that as *n* increases, there is at least a 36% probability of offloading to the best node [19].

### 4.2. Time-Optimized Multi-Task Offloading Model

In this section, the situation that arises when a mobile node wants to offload a structured task into *K* subtasks to the MEC server and perform a data analysis task on the move is investigated. Data analysis tasks can be data correlation analysis, inference and predictive analysis [27], statistical learning model building, model selection [28,29] or data from HD maps, as shown in [21]. Data can be collected through different applications such as Mobile Crowd Sensing (MCS) or Vehicle Crowd Sensing (VCS) [30,31]. The mobile node sequentially observes the MEC server deployed in the RSU and decides whether to offload or not, and once rejected or selected, it cannot be reversed. The mobile node can perform a relative ranking of the historical observed *X* values, and within the observation time *n*, a total of *K* decisions are required to ensure that all *K* blocks of tasks are offloaded.

In addition, the tasks that need to be offloaded have contextual data time limits, so it is necessary to advance the offloading as much as possible to prevent data from being obsolete and not being processed. Using the observed server serial number *i* as the variable, where *i* can be interpreted as the time of the whole offload decision process, the i=1,2,…,n. The time-optimized function *T*, adapted from [32], is constructed to consider the timeliness of the offloading decision:*T* is linear non-increasing in *i*.When i=1, the observable time starts, the value of *T* is maximum, T=1.When i=n, the observable time is over, the value of *T* is minimum.

The specific *T* function is defined as follows:(3)$$T_{i}=\left\{\begin{array}{c} \frac{n+1-i}{n},\;1\leq i\leq n.\\ 0,\;\;i>n. \end{array}\right.$$

The following two multi-tasks offloading decisions are proposed for the structured task offloading problem:1.For the general distribution scenarios of *X*, a time-optimized K-Best Choice (K-Best) strategy is proposed with the objective of maximizing the probability of selecting the best servers.2.For some realistic road scenarios where the CPU utilization may have an approximately uniform distribution, a time-optimized K-Best selection based on Uniform distribution (KBU) strategy is proposed with the goal of minimizing the total processing time for K selections.

#### 4.2.1. The K-Best Model

The offloading problem of such structured tasks can be abstracted as a K-best selection problem based on the OST; with the goal of maximizing the probability of selecting the best servers, the offloading algorithm is designed with the idea of *K* iterations. The description of K-Best model is given below: First, we define *K* stop nodes: $1\leq j_{1}<j_{2}<\ldots <j_{k}\leq n$; here, $c_{i}$ denotes the first *i* decision sequence number that has been selected. Then, we define two types of offload candidate servers, and the mobile node determines the type of candidate servers based on the relative ranking of the selected server utilization and the given threshold value θ. For example, if *m* offload selections have been made so far, the selected *m* servers are sorted in ascending order of utilization as: 1<2<…<m. The candidate server types are defined follows:(1)Compulsory candidate server α: the current server utilization detected by the mobile node is $X_{i}$; if $X_{i}<\theta$ and the relative ranking $X_{i}\leq m$, then it is a compulsory candidate server α.(2)Marginal candidate server β: the current server utilization detected by the mobile node is $X_{i}$; if $X_{i}<\theta$ or the relative ranking $X_{i}\leq m+1$, then it is a marginal candidate server β.

The Algorithm 1 is described as follows:

**Algorithm 1** The K-Best algorithm.**Require:** stop node $1\leq j_{1}<j_{2}<\ldots <j_{K}\leq n$, random variables *X*, time optimization function $T_{i}$.**Ensure:** total time delay. **for**
p=1,2,3,…,n
**do**    **for**
m=1,2,…,K
**do**     ** if**
m==1
**then**      **if**
$p\geq j_{m}$
**and**
*p* = α or β **then**        perform offload;       $j_{m}=p$;      **end if**     **else if**
m>1
**and**
p=α
**and**
$p\in (j_{m}-1,j_{m})$
**then**      perform offload;      $j_{m}=p$;     **else if**
m>1
**and** no α is in $\left(j_{m-1},j_{m}\right)$ **then**      **if**
$p\geq j_{m}$
**and**
*p* = α or β **then**       perform offload;       $j_{m}=p$;      **end if**     **end if**    **end for**
 **end for**

According to [33], it is difficult to give the general closed forms of the stopping nodes when K>1, but the structure of the process of calculating the solution and the asymptotic form of the solution can be given.

With the consideration of time-optimized function *T*, taking K=2 as an example, there are two possible decisions:

$c_{1}<j_{1}\leq c_{2}$ or $c_{1}<c_{2}\leq j_{2}$, the probability of the first case is:(4)$$\begin{array}{cc} P_{1}\left(j_{2}\right) & =\sum_{c_{2}=j_{2}}^{n}\prod_{k=c_{1}+1}^{j_{2}-1}T_{k}\left(1-\frac{1}{k}\right)\prod_{l=j_{2}}^{c_{2}-1}T_{l}\left(1-\frac{1}{l}\right)\frac{2}{c_{2}}\prod_{s=c_{2}+1}^{n}T_{s}\left(1-\frac{2}{s}\right)\\ \; & =\sum_{c_{2}=j_{2}}^{n}\prod_{i=c_{1}+1}^{n}T_{i}\frac{c_{1}}{n}\frac{2}{c_{2}-2}\frac{j_{2}-2}{n-1}. \end{array}$$

Let $P_{1}\left(j_{2}\right)=\frac{c_{1}\left(n-c_{1}\right)!}{n^{n-c_{1}}}F_{1}\left(c_{1},j_{2}\right)$, the probability of the second case is:(5)$$\begin{array}{cc} P_{2}\left(j_{2}\right) & =\sum_{c_{2}=c_{1}+1}^{j_{2}-1}\prod_{k=c_{1}+1}^{c_{2}-1}T_{k}\left(1-\frac{1}{k}\right)\frac{1}{c_{2}}T_{c_{2}}\prod_{l=c_{2}+1}^{n}T_{l}\left(1-\frac{2}{l}\right)\\ \; & =\sum_{c_{2}=c_{1}}^{j_{2}-1}\prod_{i=c_{1}+1}^{n}T_{i}\frac{c_{1}}{n\left(n-1\right)}. \end{array}$$

Let $P_{2}\left(j_{2}\right)=\frac{c_{1}\left(n-c_{1}\right)!}{n^{\left(n-c_{1}\right)}}F_{2}\left(c_{1},j_{2}\right)$, the total probability can be expressed as:(6)$$\begin{array}{c} P\left(j_{2}\right)=P_{1}\left(j_{2}\right)+P_{2}\left(j_{2}\right). \end{array}$$

When n→∞, we use the Euler–McLaughlin formula to replace the sum of probabilities; then, the total probability can be expressed as:(7)$$\begin{array}{cc} V_{1}\left(c_{1},j_{2}\right) & =\frac{c_{1}\left(n-c_{1}\right)!}{n^{\left(n-c_{1}\right)}}\left(\int_{j_{2}}^{1}2\frac{j_{1}}{c_{2}}dc_{2}+\int_{c_{1}}^{j_{2}}1dc_{2}\right)\\ \; & =\frac{c_{1}\left(n-c_{1}\right)!}{n^{\left(n-c_{1}\right)}}\left(-c_{1}+j_{2}-2j_{2}ln\left(j_{2}\right)\right). \end{array}$$

Since the function $V_{1}\left(c_{1},j_{2}\right)$ is unimodal in $j_{2}$, the maximum *P* value is given when $\frac{dV_{1}}{dj_{2}}=0$:(8)$$\begin{array}{cc} \frac{dV_{1}}{dj_{2}} & =c_{1}\left(1-2lnj_{2}-2\right)=0, \end{array}$$
(9)$$\begin{array}{c} j_{2}^{*}=e^{-\frac{1}{2}}=0.6065306596\ldots \end{array}$$

Next, we find the value of $j_{1}$; again, there are two cases: $j_{2}=j_{2}^{*}\left(ifc_{1}<j_{2}^{*}\right)$ or $j_{2}=c_{1}\left(ifc_{1}\geq j_{2}^{*}\right)$; the probability of the first case is:(10)$$\begin{array}{cc} P_{1}\left(j_{1}\right) & =\sum_{c_{1}=j_{1}}^{j_{2}^{*}-1}\prod_{i=j_{1}}^{c_{1}-1}T_{i}\left(1-\frac{1}{i}\right)\frac{1}{c_{1}}c_{2}\left[F_{1}\left(c_{1},j_{2}^{*}\right)+F_{2}\left(c_{1},j_{2}^{*}\right)\right]\\ \; & =\sum_{c_{1}=j_{1}}^{j_{2}^{*}-1}\frac{\prod_{i=j_{1}}^{c_{1}-1}n+1-i}{n^{\left(c_{1}-j_{1}+2\right)}}\frac{j_{1}-1}{c_{1}-1}\left[F_{1}\left(c_{1},j_{2}^{*}\right)+F_{2}\left(c_{1},j_{2}^{*}\right)\right]. \end{array}$$

$P_{1}\left(j_{1}\right)$ is unimodal in $j_{1}$; the probability of the second case is:(11)$$\begin{array}{cc} P_{2}\left(j_{1}\right) & =\sum_{c_{1}=j_{2}^{*}}^{n-1}\sum_{c_{2}=c_{1}+1}^{n}\prod_{i=j_{1}}^{c_{1}-1}T_{i}\left(1-\frac{1}{i}\right)\frac{1}{c_{1}}T_{c_{1}}\prod_{l=c_{1}+1}^{c_{2}-1}\left(1-\frac{2}{l}\right)\frac{2}{c_{2}}T_{c_{2}}\prod_{s=c_{2}+1}^{n}\left(1-\frac{2}{s}\right)\\ \; & =\frac{(n+1-j_{1})!}{n^{\left(n+1-j_{1}\right)}}\sum_{c_{1}=j_{2}^{*}}^{n-1}\sum_{c_{2}=c_{1}+1}^{n}\frac{2}{c_{2}-2}\frac{j_{1}-1}{n}\frac{1}{n-2}, \end{array}$$

$P_{2}\left(j_{1}\right)$ is monotonic in $j_{1}$; the total probability can be expressed as:(12)$$\begin{array}{c} P\left(j_{2}\right)=P_{1}\left(j_{2}\right)+P_{2}\left(j_{2}\right). \end{array}$$

Because $P_{1}\left(j_{1}\right)$ is unimodal in $j_{1}$, $P_{2}\left(j_{1}\right)$ is monotonic in $j_{1}$, so $P(j_{1})$ is unimodal in $j_{1}$. Similarly, when n→∞, we use the Euler–Maclaurin formula to replace the sum of probabilities; then, the total probability can be expressed as:(13)$$\begin{array}{c} V_{2}\left(j_{1}\right)=\int_{j_{1}}^{j_{2}^{*}}\frac{1}{c_{1}}V_{1}\left(c_{1},j_{2}^{*}\right)dc_{1}+j_{1}\int_{c_{1}=j_{2}^{*}}^{1}\int_{c_{2}=c_{1}}^{1}\frac{2}{c_{2}}dc_{2}dc_{1}. \end{array}$$

Since $V_{2}\left(j_{1}\right)$ is unimodal in $j_{1}$, the maximum *P* value is given when $\frac{dV_{2}}{dj_{1}}=0$:(14)$$\begin{array}{c} j_{1}^{*}=e^{-\frac{1}{2}}W(-e^{\left(-3+e^{\frac{1}{2}}\right)})=0.2291147286\ldots . \end{array}$$Here, W(x) is the Lambert function. Based on a similar solution structure, we can derive in advance the form of the asymptotic solution for the stop nodes at K=3 for the subsequent experiments:(15)$$\begin{array}{cc} \frac{j_{1}}{n} & \to 0.1666171752\ldots .\\ \frac{j_{2}}{n} & \to -e^{-\frac{1}{3}}W\left(-exp\left(-\frac{2}{5}+e^{\frac{1}{3}}\right)\right)=0.4369818602\ldots .\\ \frac{j_{3}}{n} & \to 0.7165313106\ldots . \end{array}$$

The stop nodes can be calculated in advance and stored in the initialization parameters of the model without delaying the unloading. The space complexity of the K-Best algorithm is S(n)=O(n) and the time complexity is T(n)=O(n).

#### 4.2.2. The KBU Model

In some realistic road scenarios, the distribution of utilization *X* of edge servers deployed in the RSU may be closer to a uniform distribution. In this case, the threshold corresponding to each observation node can be bounded in reverse by using the function of uniform distribution [34]. We propose the KBU model with the objective of minimizing the total *K* processing delays, and the thresholds are bounded by the time optimization function $T_{i}$ at the same time.

In this model, nodes are required to select online *K* servers, and in order to tolerate a certain rate of offloading failures, we define a parameter $K^{{}^{\prime }}$ that represents the expectation of the number of global offloads. $K^{{}^{\prime }}$ does not have to be an integer, $K^{{}^{\prime }}\geq K$. A set of metrics $I_{1},I_{2},\ldots ,I_{n}$ is defined, which indicate whether the *i*th server is selected or not. $I_{i}=0$ indicates that the *i*th server is not selected, and $I_{i}=1$ shows the opposite, while $\delta_{n}$ denotes the set of all $I_{i}$ indicators.

*K* and $K^{{}^{\prime }}$ satisfy the following two constraints:

Constraint 1:(16)$$\begin{array}{c} \sum_{i=1}^{n}I_{i}\geq K,K\in N,0\leq K\leq n. \end{array}$$

Constraint 2:(17)$$\begin{array}{c} E\left(\sum_{i=1}^{n}I_{i}\right)\geq K^{{}^{\prime }},K^{{}^{\prime }}\in R,0\leq K^{{}^{\prime }}\leq n. \end{array}$$

The total time delay under different constraints is denoted by $V_{K,K^{{}^{\prime }}}\left(n\right)$; the objective of the strategy is to minimize the total time delay $V_{K,K^{{}^{\prime }}}\left(n\right)$:(18)$$\begin{array}{c} V_{K,K^{{}^{\prime }}}\left(n\right)=\min_{\left(I_{1},\ldots ,I_{n}\right)\in \Gamma_{n}}E\left(\sum_{i=1}^{n}I_{i}X_{i}\right),n\geq K^{{}^{\prime }}\geq K,r,n\in N. \end{array}$$

$H_{k}=\left(X_{1},I_{1}\right),\left(X_{2},I_{2}\right)\ldots ,\left(X_{k},I_{k}\right)$ is the offload history, there are $N_{k}$ tasks that have been offloaded, 0≤N−k≤K; then, there is the structure of the basic solution as:(19)$$\begin{array}{c} V_{K,K^{{}^{\prime }}}\left(n\right)=V_{K-N_{k},K^{{}^{\prime }}-N_{k}}\left(n-k\right)+\sum_{j=1}^{k}I_{j}X_{j}. \end{array}$$

Let δ be the minimum time to satisfy constraint 1, so that:(20)$$\begin{array}{c} V_{K,K^{{}^{\prime }}}\left(n\right)=V_{0,K^{{}^{\prime }}-K}\left(n-\delta \right)+\sum_{j=1}^{\delta }I_{j}X_{j}. \end{array}$$

Among them:(21)$$\begin{array}{c} V_{0,K^{{}^{\prime }}-K}\left(i\right)=\frac{{\left(K^{{}^{\prime }}-K\right)}^{2}}{2i}. \end{array}$$

Now, we can get the double recursive solution form of $V_{K,K^{{}^{\prime }}}\left(n\right)$:(22)$$\begin{array}{c} V_{K,K^{{}^{\prime }}}\left(n\right)=V_{K,K^{{}^{\prime }}}\left(n-1\right)-\frac{1}{2}{\left[V_{K,K^{{}^{\prime }}}\left(n-1\right)-V_{K-1,K^{{}^{\prime }}-1}\left(n-1\right)\right]}^{2}. \end{array}$$

The optimal threshold $t^{*}\left(\rho ,k\right)$ denotes the threshold value for the *k*th server detected in the offload decision for the *ρ*th task. $t^{*}\left(\rho ,k\right)$ can be given by the inverse of the structure of the optimal total time cost $V_{K,K^{{}^{\prime }}}\left(n\right)$:(23)$$\begin{array}{cc} t^{*}\left(\rho ,k\right) & =t_{K,K^{{}^{\prime }}}^{*}\left(\rho ,k\right)\\ \; & =V_{K-\rho ,K^{{}^{\prime }}-\rho }\left(k-1\right)-V_{K-\rho -1,K^{{}^{\prime }}-\rho -1}\left(k-1\right). \end{array}$$

That is, for the first *ρ*th task, if the currently observed $X_{i}\leq t^{*}\left(\rho ,n-k\right)$, the server *i* should be taken into account.

Considering that offloading should be performed as far as possible before the task data become obsolete, the time-optimized function is further used to constrain the threshold to obtain a new threshold defined as t(ρ,k):(24)$$\begin{array}{c} t\left(\rho ,k\right)=t^{*}\left(\rho ,k\right)*T_{\left(n-k\right)}. \end{array}$$

For example, when K=3, n=10, the KBU model gives different thresholds for the three subtasks, and a comparison of t(ρ,k) and $t^{*}\left(\rho ,k\right)$ before and after optimization is shown in Figure 3.

The Algorithm 2 is given as follows:

**Algorithm 2** The KBU algorithm.**Require:** distribution of random variable *X*, time-optimized function $T_{i}$.**Ensure:** total time delay $V_{K,K^{{}^{\prime }}}\left(n\right)$. **A Obtain the minimum time delay:** (A1) **for**
$i=\left\lceil K^{{}^{\prime }}\right\rceil -K,\left\lceil K^{{}^{\prime }}\right\rceil -K+1,\ldots ,n$
**do**   $V_{0,K^{{}^{\prime }}-K}\left(i\right)={\left(K^{{}^{\prime }}-K\right)}^{2}/\left(2i\right)$; **end for** (A2) **for**
$i=\left\lceil K^{{}^{\prime }}\right\rceil -K,\left\lceil K^{{}^{\prime }}\right\rceil -K+1,\ldots ,n-k$;s=1,2,…,k
**do**   $V_{s,i}\left(i\right)=i/2$; **end for** (A3)  **for**
s=1,2,…,K and the initial conditions (A1) and (A2) **do**   $V_{s,K^{{}^{\prime }}-K+s}\left(i\right)=K^{{}^{\prime }}-K+s,\ldots ,n-K+1,n-K,\ldots ,n$; **end for** **B Obtain the optimal threshold:** (B1) **for**
$i=\left\lceil K^{{}^{\prime }}\right\rceil -K,\left\lceil K^{{}^{\prime }}\right\rceil -K+1,\ldots ,n+K$
**do**   $t\left(K,i\right)=V_{0,K^{{}^{\prime }}-K}\left(i\right)={\left(K^{{}^{\prime }}-K\right)}^{2}/\left(2i\right)$; **end for** (B2) **for**
ρ=0,1,…,K−1
**do**   **for**
i=K−ρ,K−ρ+1,…,n
**do**    $t^{*}\left(\rho ,i\right)=V_{K-\rho ,K^{{}^{\prime }}-\rho }\left(i-1\right)-V_{K-\rho -1,K^{{}^{\prime }}-\rho -1}\left(i-1\right)$;   **end for** **end for** $t\left(\rho ,k\right)=V^{*}\left(\rho ,k\right)V_{n-k}$; **for**
p=1,2,3,…,n
**do**   **for**
m=1,2,3,…,K−1
**do**    **if** number of nodes left ≤ number of unloaded tasks left **then**      perform offload;    **else if**
$X_{p}<t\left(m,n-p\right)$
**then**      perform offload;    **end if**   **end for** **end for**

Same as the K-Best model, the *K* thresholds here can be calculated in advance and stored in the initialization parameters of the model without delaying the unloading. The space complexity of the KBU algorithm is S(n)=O(n), and the time complexity is T(n)=O(n).

## 5. Experimental Evaluation

We use two settings to evaluate the two proposed OST-based multi-task offloading models: simulation-based evaluation and real-world data sets based. In both settings, we compare our models: namely, K-Best (Section 4.2.1) and KBU (Section 4.2.2) with the Optimal model [19], BCP model [17], Random selection model (Random) and the *p*-stochastic model (*p*-model). We will discuss these models next.

The four offloading models used for comparison in a multitask offloading scenario are described as follows:

The optimal model takes the number of observations *n*, the probability distribution function of *X*, the quality-aware function and the threshold θ as inputs, and outputs the index *s* where nodes start checking the MEC servers. If the subsequent observed *X* does not exceed θ, i.e., X≤θ, then we perform offloading; otherwise, we continue to observe. When the number of remaining unselected servers *m* does not exceed the number of remaining unloaded tasks, i.e., $m\leq K-K_{p}$, then the remaining tasks are offloaded to the corresponding servers sequentially without comparing so that all tasks are guaranteed to be offloaded.

The BCP model takes the number of observations *n* and the probability distribution function of *X* as inputs and calculates the stop node $r_{n}$, rejects the first $r_{n}-1$ servers, and then selects the first server with the best relative previous ranking (if any) to offload the task. When there are $K_{p}$ tasks that have been offloaded, there are still $K-K_{p}$ tasks left to be offloaded. If the number of remaining unselected servers *m* does not exceed the number of remaining unloaded tasks, i.e., $m\leq K-K_{p}$, then the remaining tasks are offloaded to the corresponding servers in turn to ensure that all tasks can be offloaded.

In the *p*-model model, we use p=0.8 as the offload probability for each server in the following experiments, i.e., the node connects to the MEC server during the move and has an 80% probability of offloading to the server it is currently observing. In the random model, the mobile node randomly selects *K* servers for task offloading, and again, when the number of remaining unselected servers *m* does not exceed the number of remaining unloaded tasks, i.e., $m\leq K-K_{p}$, then the remaining tasks are offloaded to the corresponding servers sequentially without further observation, so that all tasks are guaranteed to be offloaded.

To simulate the MEC environment, we use Simpy [35] in Python. Simpy is a process-based framework for discrete-event simulation. Each MEC server is modeled as a resource, and during the simulation, the server publishes its processing time *X*, and the connected mobile node can receive the information. The mobile node is modeled as a process that detects the processing time of the server in a one-direction mobile model and chooses whether to offload it or not. The values of the simulation experiment are shown in Table 2.

### 5.1. Evaluation Based on Simulated Data

In the evaluation based on simulated data, the node would know in advance the distribution function of the random variable *X*, i.e., server utilization or CPU utilization. We consider experiments in the case of K=3, n=10. A series of random numbers will be generated by Python functions, which obey a specific distribution, such as normal or uniform distribution. Consider first the case where *X* follows a normal distribution.

As shown in Figure 4, the results between the K-Best model and the four comparison models are compared when the distribution of *X* presents a case of normal distribution. The K-Best model clearly performs best in terms of total delay of offloading. We can observe a clear overlap between the K-best model and Optimal model in Figure 4a, and this overlap is significantly reduced in the comparison plots with the other three models. It can be seen that the K-Best model works significantly better than the last three models and presents Optimal. Overall, the OST-based models (K-Best, Optimal and BCP) are significantly more effective than several other models that are not based on OST. It can be seen in Figure 5 that the OST-based models achieve a lower expected total processing delay, and the K-Best model has similar results to the Optimal and BCP models, while it differs more from the *p*-model and random model.

In the previous experiments, the random variable *X* observed by the mobile nodes followed a normal distribution. Now, let *X* be uniformly distributed scaled in [0,1]; *X* is the server utilization or CPU utilization. For example, X=0.5 means that the CPU utilization of the server is 50%. For all models, the following experiments follow the same steps as in the previous experiments.

As shown in Figure 6, the KBU model clearly performs the best compared to the remaining four models, and it can be seen in Figure 6a that the KBU model overlaps significantly with the K-Best and Optimal models. Overall, the four OST-based models perform better than the other models. The KBU model achieves the smallest expected processing delay with *X* following a uniform distribution, and the K-Best model also achieves a more desirable expected processing delay, which is significantly better than the *p*-model and the Random model, as can be seen in Figure 7.

#### Sensitivity Analysis

As shown in Figure 8, the total processing delay of the K-Best model tends to decrease as the number of servers increases with a constant number of tasks K=3. Figure 8a shows the case when the random variable *X* obeys a normal distribution, and the case when the random variable *X* obeys a uniform distribution is shown in Figure 8b. We can see that when the number of observable servers increases, the K-Best model allows mobile nodes to select the best possible server for task offloading and has general applicability to the distribution of *X*.

As shown in Figure 9, as the number of servers increases, the total processing latency of the KBU model tends to decrease with a constant number of tasks K=3. It means that the KBU model enables mobile nodes to select the best possible server for task offloading as the number of observable servers increases.

### 5.2. Evaluation Based on Real Data

We also consider real data sets to evaluate our models. The purpose of this evaluation is to see how our models perform when dealing with real data sets. We use the CABS data set provided by the Shenzhen Smart City project [36] to simulate the movements of the mobile nodes. The data set contains four attributes of the vehicles: vehicle ID, GPS location, movement time and movement speed. The use of mobility trace here is not for studying the mobility of users; in our experiment, for each movement, the car picks a server from the servers’ data set, checks that server utilization, and makes a decision of whether the car should offload at that time or continue observing based on the decision suggested by the models, as explained earlier in the Simulation Evaluation section. An example of one connection observation is shown in Table 3; we can see that Cab 178 made connections to three servers at different times, where two connections were made to server m_1938 at almost the same location but at different times, while obtaining server information, i.e., CPU utilization.

The processing time of the servers is provided by the actual CPU utilization obtained in the Alibaba Cluster Tracking Program [37]. There are more than one billion rows of CPU utilization data from about 150 servers that are recorded in the data set. One million of these data are used in the experiment; Figure 10 shows the probability distribution of the CPU utilization of all servers in the data set. We can see in Figure 10 that the CPU utilization follows a normal distribution with μ=36, σ=16. The values of the key parameters’ values in the experiment are given in Table 4.

At the beginning of the experiment, the mean and the standard deviation were taken once for the whole servers’ utilization data set to feed the models (K-Best, KBU, Optimal and BCP). In a realistic scenario, the mobile nodes do not know the mean and standard deviation of a particular MEC server, but they can obtain the information about the historical data of the MEC servers in one area in a specific time with the help of MEC servers operators. Therefore, we take this information once when we start the experiment.

Similar to the experiments based on simulated data, the results of each model are aggregated and compared in the experiments based on real data sets. Each model selects *K* servers for mobile nodes to offload to minimize the total offload delay. The total delay is expressed in terms of the total server utilization.

Figure 11 shows the average server utilization for the offloading decisions suggested by each model. We can see that in the results based on real data sets, the OST-based model still performs better at minimizing the total offloading delay, and the KBU model and the K-Best model perform the best. Since in real data sets, the distribution of CPU utilization is closer to a normal distribution than a uniform distribution, the KBU model fails to show a more significant advantage over the K-Best model.

Figure 12 shows the average waiting time for the offloading decision suggested by each model. We can see that the *p*-model with p=0.8 suggests the smallest average waiting time. However, the average offloading time is too long, so that choosing an instant server is not a wise offload strategy. Moreover, the optimal server is unknown, and the mobile node cannot know exactly which server is the best choice, so using the OST-based model can achieve a near-optimal server utilization.

#### Sensitivity Analysis

We use the number of successful offloading for different threshold requirements as a performance metric for the models. The number of successful offloading is the number of offload decisions suggested by each model that satisfy a specific requirement. Suppose we have three different MEC applications x, y, and z, all with specific requirements. For example, application x requires a total CPU utilization ≤0.4, application y requires a total CPU utilization ≤0.6, and application z requires a total CPU utilization ≤0.8.

Figure 13 shows the number of successful offloading for all models for different requirements. For the first case requiring a total CPU utilization ≤0.4, the KBU model achieves 89 successful offloads, and the K-Best model achieves 90 successful offloads; for the second case where the required total CPU utilization ≤0.6, the KBU model achieves 250 successful offloads and the K-Best model achieves 230 successful offloads; for the third case that the total CPU utilization should ≤0.4, the KBU model achieves 385 successful offloads, and the K-Best model achieves 408 successful offloads. Overall, we can see that the OST-based offloading models (KBU, K-Best and Optimal) are significantly more effective than the other models.

## 6. Conclusions

In this paper, we focus on a time-optimized multi-tasking offloading model based on OST in an IoV environment. A time-optimized multi-task offloading model adapted to a general *X* distribution and a time-optimized multi-task offloading model that performs better for the case of uniform *X* distribution are proposed, and detailed evaluations are provided. The experimental evaluations show that the OST-based models outperform other offloading methods, achieve more desirable expected processing latencies, are efficient to use in mobile nodes, and do not require significant resources. In addition, the OST-based models are suitable for scenarios where mobile nodes need to make local and independent decisions in the MEC environment. In realistic scenarios, mobile nodes can obtain the required information with the help of MEC service providers and thus can make decisions autonomously. As future work, it is considered to study the K-Best model when *K* keeps increasing and may give a feasible closed-form solution. Since the observed recall of the MEC server is allowed in different mobility models, the optimization of the mult-itasking offloading model based on OST theory can be further investigated in the recallable server as well as in the multi-node competition scenario.

## Figures and Tables

**Figure 1 entropy-24-00814-f001:**
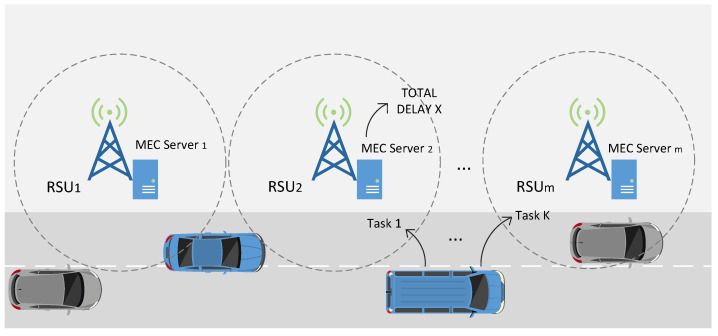
MEC in the Internet of Vehicles.

**Figure 2 entropy-24-00814-f002:**
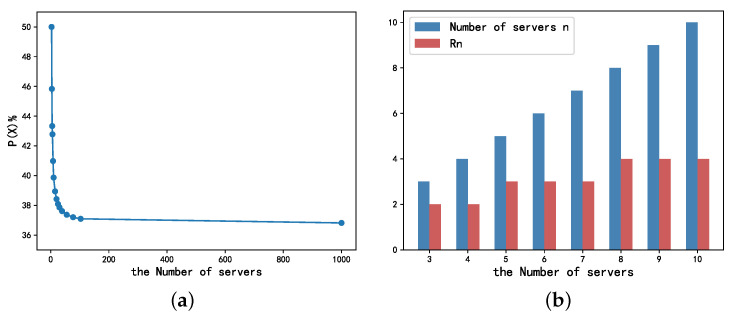
The probability of offloading to the best (**a**) and the value of $R_{n}$ (**b**) for different numbers of servers *n*.

**Figure 3 entropy-24-00814-f003:**
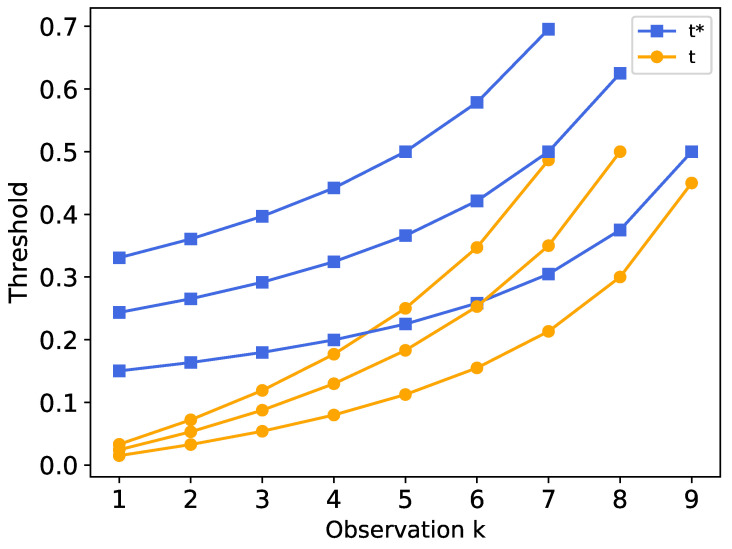
Comparison of the threshold *t* before and after the time-optimized function constraint.

**Figure 4 entropy-24-00814-f004:**
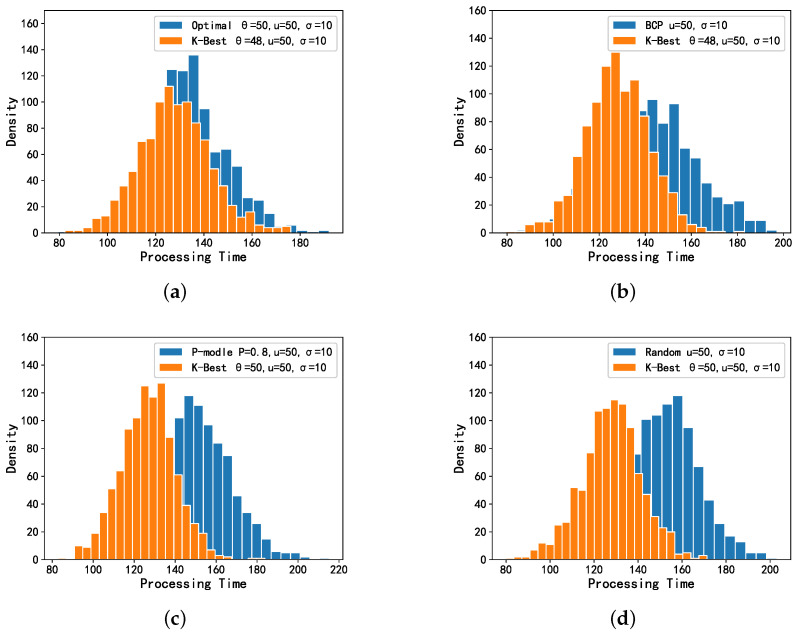
Simulation results for all the models when *X* is normally distributed. (**a**) Optimal and K-Best selections. (**b**) BCP and K-Best selections. (**c**) P-model and K-Best selections. (**d**) Random and K-Best selections.

**Figure 5 entropy-24-00814-f005:**
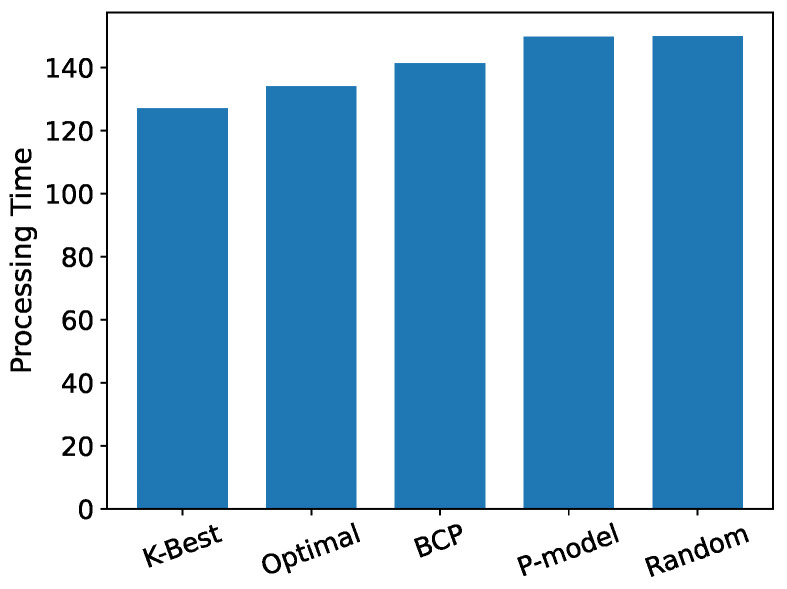
Confidence interval of simulation results for each model when *X* is normally distributed.

**Figure 6 entropy-24-00814-f006:**
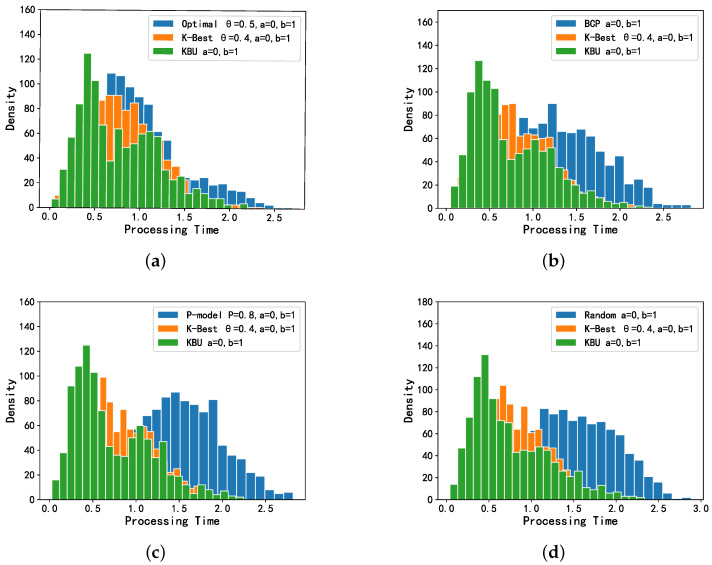
Simulation results for all the models when *X* is uniformly distributed. (**a**) Optimal, K-Best and KBU selections. (**b**) BCP, K-Best and KBU selections. (**c**) P-model, K-Best and KBU selections. (**d**) Random, K-Best and KBU selections.

**Figure 7 entropy-24-00814-f007:**
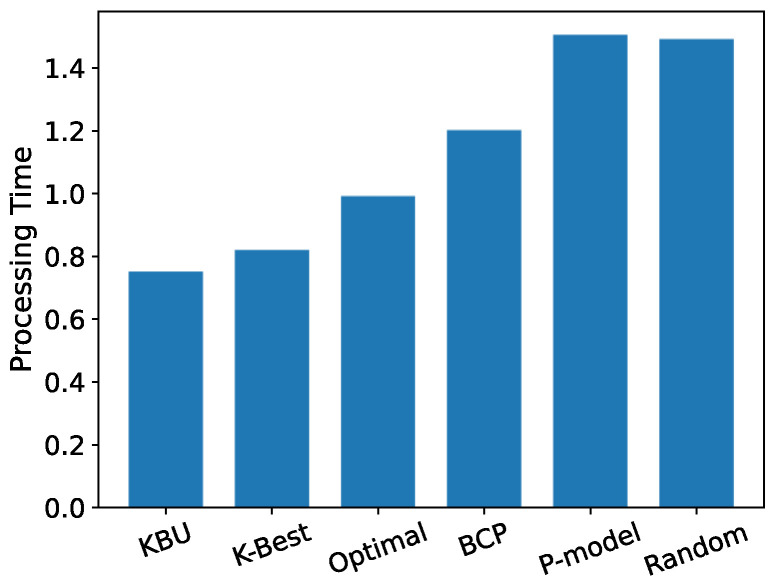
Confidence interval of simulation results for each model when *X* is uniformly distributed.

**Figure 8 entropy-24-00814-f008:**
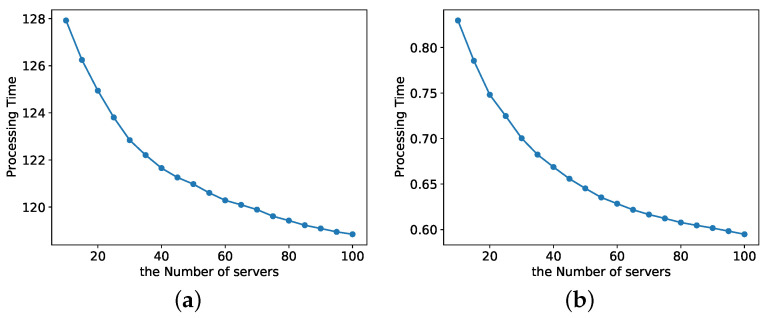
Total latency of the K-Best model varies with the number of servers *n*. (**a**) *X* normally dstributed. (**b**) *X* uniformly distributed.

**Figure 9 entropy-24-00814-f009:**
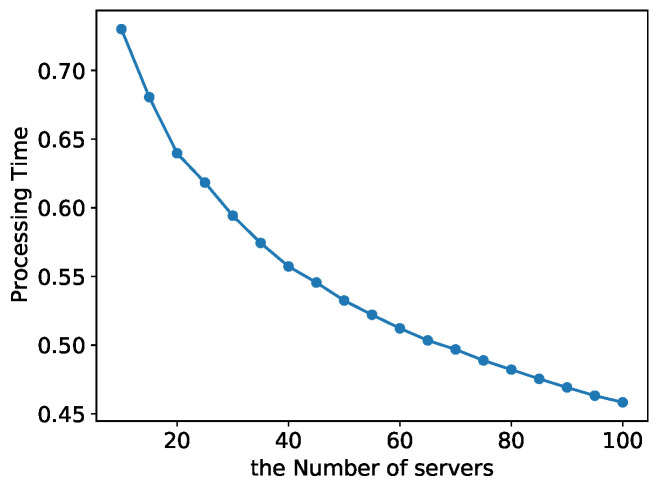
Total latency of the KBU model varies with the number of servers *n*.

**Figure 10 entropy-24-00814-f010:**
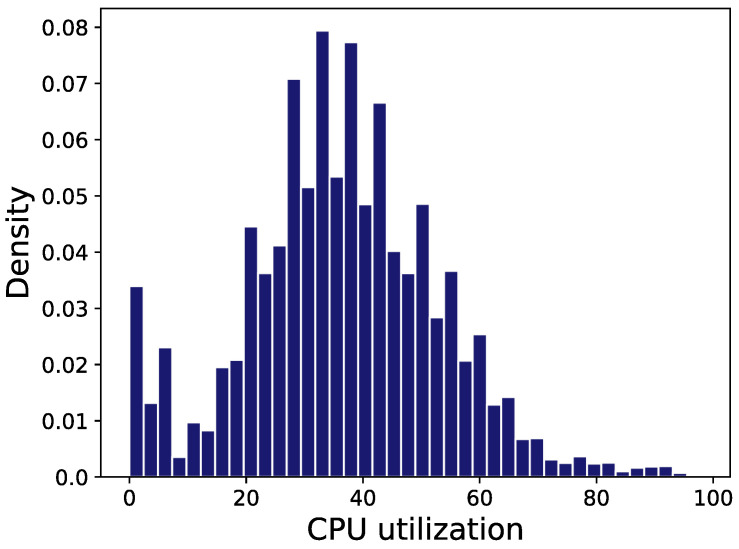
The distribution of the servers’ CPU utilization.

**Figure 11 entropy-24-00814-f011:**
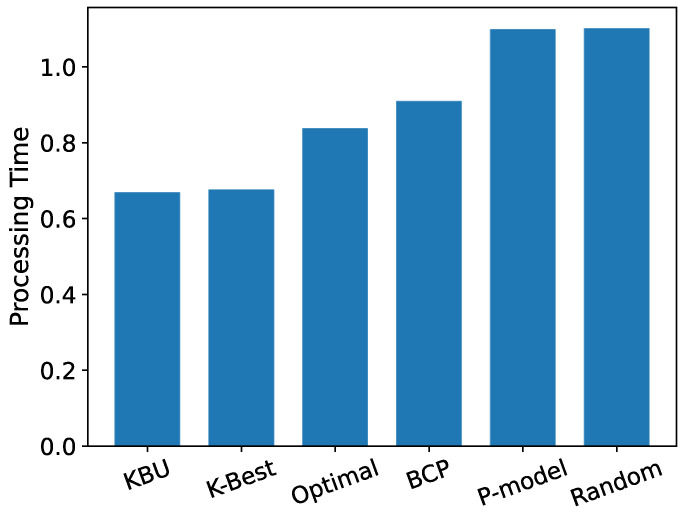
Average offloading time.

**Figure 12 entropy-24-00814-f012:**
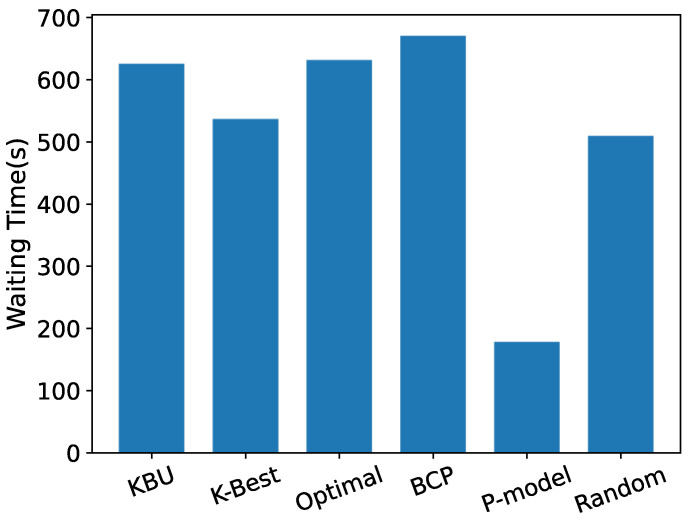
Average waiting time.

**Figure 13 entropy-24-00814-f013:**
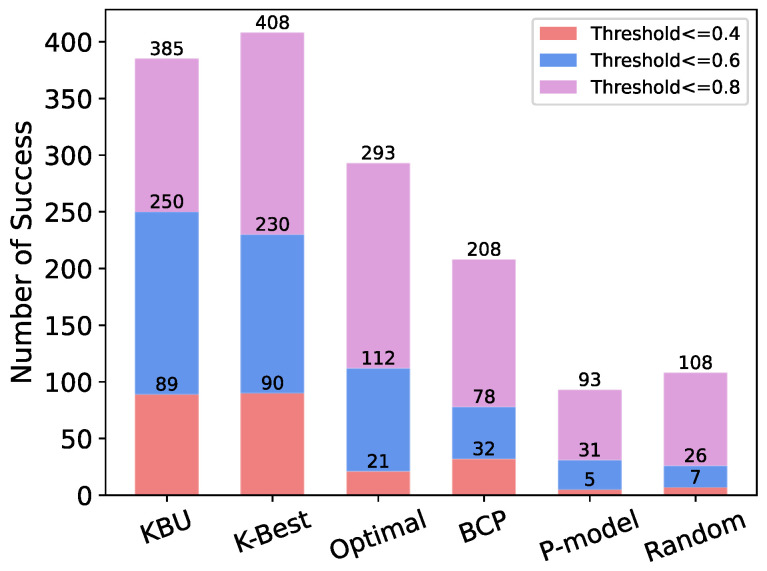
The number of successful offloadings for each model based on different threshold values.

**Table 1 entropy-24-00814-t001:** Key notations used in the paper.

Notation	Explanation
*X*	Random variable for server processing time
*i*	Random variable for server processing time
*n*	The serial number of the observed server
θ	Threshold in the offloading model
*K*	Number of tasks that need to offload
$r_{n}$	Stop node in the BCP model
$P_{n}$	Maximum probability in the BCP model
$T_{i}$	Time-optimized function
$c_{i}$	The *i*-th decision number that has been selected
$P_{i}\left(j_{k}\right)$	The probability of the *i*-th possible offloading branch in the K-Best model
α	Mandatory candidate in the K-Best model
β	Edge candidate in the K-Best model
$j_{k}$	Stop nodes in the K-Best model
$V_{k,k^{{}^{\prime }}}\left(n\right)$	Total time delay in KBU model
$t_{(}\rho ,k)$	Optimal threshold after time optimization in KBU model
*P*	The probability *P* in the p-model

**Table 2 entropy-24-00814-t002:** Simulation experiment parameters’ values.

Parameters	Value/Rang
*X*	*N* (50,10), *U* (0,1)
Number of mobile nodes	1000
*K*	3
*n*	10
Threshold θ of K-Best	48
Threshold θ of Optimal	50
*P* for the *p*-modle	0.8

**Table 3 entropy-24-00814-t003:** A sample of the data set used in the experiment.

Taxi ID	Movement ID	Location	Machine Name	CPU Utilization
178	2014/10/22 8:00:20	(22.5965, 114.114601)	m_1935	(23)
178	2014/10/22 8:00:50	(22.5966, 114.1182 02)	m_1937	(35)
178	2014/10/22 8:01:20	(22.5968, 114.1194)	m_1938	(54)
178	2014/10/22 8:01:50	(22.5965, 114.119598)	m_1938	(26)

**Table 4 entropy-24-00814-t004:** Real data set experiment parameters’ values.

Taxi ID	Movement ID
*X*	Real servers CPU utilization in N(36,16)
Number of mobile nodes	1000
*K*	3
*n*	10
Threshold θ of K-Best θ	48
Threshold θ of Optimal θ	50
*P* for the *p*-modle	0.8

## Data Availability

Data is contained within the article.

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
