# Peer review of "Multi-Task Offloading Based on Optimal Stopping Theory in Edge Computing Empowered Internet of Vehicles"

_entropy, 2022, doi:10.3390/e24060814_

Round 1

Reviewer 1 Report

This paper proposed a time-optimized, multi-task-offloading model adopting the principles of optimal stopping theory with the objective of maximizing the probability of offloading to the optimal servers. The paper also proposed another OST-based model with the objective of minimizing the floading delay. The proposed models are experimentally compared and evaluated with related OST models using simulated data sets and real data sets, and sensitivity analysis is performed. The experimental evaluations show that the OST-based models outperform other  offloading methods, achieve more desirable expected processing latencies, are efficient to  use in mobile nodes, and do not require significant resources.

The paper is well-written. The paper is technically sound and has shown comprehensive experimental. The contributions are clear. Good work. Just one question.

What about complexity analysis of the technique and mathematical proof?

Author Response

We have included the response letter and the revised manuscript in the attachment.

Reviewer 2 Report

The manuscript considers a time-optimized multitasking offloading model based on optimal stopping theory in an IoV environment. The proposed algorithms use time-optimized multi-task offloading models adapted to a general and uniform distribution. Appropriate theoretical framework was applied. The simulation results confirm that the proposed methods allow achieving a higher result than existing schemes. Nonetheless, there are aspects that require improvements:

1) What was the key motivation behind focusing on this problem? It is necessary to make more acute motivation of this paper.

2) Authors need to add the paper structure at the end of the first section.

3) Authors need to add Ref. 19 in section 4.1.

4) Correct designation r-1 instead of Rn in Fig. 2(b).

5) In my opinion, a not entirely successful example is given with a dataset from UrbanCPS [36] in section 5.2. Needs to be clarified table 3.

Author Response

(The authors gave the same response as above.)
